# Fine-grained Differentiable Physics: A Yarn-level Model for Fabrics

**Deshan Gong[1], Zhanxing Zhu[2,3], Andrew J. Bulpitt[1] and He Wang[1]**[*]
[1]School of Computing, University of Leeds
[2]School of Informatics, University of Edinburgh
[3]Peking University
{scdg, A.J.Bulpitt, h.e.wang}@leeds.ac.uk, zhanxing.zhu@pku.edu.cn

## Abstract

Differentiable physics modeling combines physics models with gradient-based learning to provide model explicability and data efficiency. It has been used to learn dynamics, solve inverse problems and facilitate design, and is at its inception of impact. Current successes have concentrated on general physics models such as rigid bodies, deformable sheets, etc, assuming relatively simple structures and forces. Their granularity is intrinsically coarse and therefore incapable of modelling complex physical phenomena. Fine-grained models are still to be developed to incorporate sophisticated material structures and force interactions with gradient-based learning. Following this motivation, we propose a new differentiable fabrics model for composite materials such as cloths, where we dive into the granularity of yarns and model individual yarn physics and yarn-to-yarn interactions. To this end, we propose several differentiable forces, whose counterparts in empirical physics are indifferentiable, to facilitate gradient-based learning. These forces, albeit applied to cloths, are ubiquitous in various physical systems. Through comprehensive evaluation and comparison, we demonstrate our model's *explicability* in learning meaningful physical parameters, *versatility* in incorporating complex physical structures and heterogeneous materials, *data-efficiency* in learning, and *high-fidelity* in capturing subtle dynamics. Code is available in: https://github.com/realcrane/Fine-grained-Differentiable-Physics-A-Yarn-level-Model-for-Fabrics.git

## 1 Introduction

Differentiable physics models (DPMs) have recently spiked interests, e.g. rigid bodies (Heiden et al., 2020), cloth (Liang et al., 2019), and soft bodies (Hu et al., 2019). The essence of DPMs is making physics models differentiable, so that gradient-based learning can be used to make systems adhere strictly to physical dynamics. This is realized via back-propagation through a series of observed actions, where the system can quickly learn the underlying dynamics. While enjoying neural networks' capability of modeling arbitrary non-linearity, DPMs also improve the model explicability as the learnable model parameters bear physical meanings. As a result, such models provide a new avenue for many applications such as inverse problems, e.g. estimating the mass of a moving rigid body (de Avila Belbute-Peres et al., 2018b), and control, e.g. learning to shake a bottle to shape the fluid in it (Li et al., 2019).

Early research attempted to model simple and general physical systems such as rigid bodies (de Avila Belbute-Peres et al., 2018b), followed by a range of systems including deformable objects (Li et al., 2019), cloth (Liang et al., 2019), contacts (Zhong et al., 2021), etc. However, existing models are only generally-purposed which do not consider complex structures/topologies and force interactions. Taking cloth (i.e. fabrics) as an example, existing models (Liang et al., 2019; Li et al., 2019) can learn general cloth dynamics, but only when the cloth is relative simple and homogeneous. Recent research (Wang et al., 2020) has started to explore articulated systems but the model capacity is insufficient to capture the full dynamics of complex systems such as fabrics. Since real-world physical

---

[*]corresponding author

systems (e.g. materials in engineering) often have sophisticated structures and consist of heterogeneous materials, we argue that it is crucial to design fine-grained DPMs, for differentiable physics to be truly applicable and meaningful to real-world applications.

This paper focuses on fabrics which are composite materials consisting of basic slim units arranged in different patterns. A common example in fabrics is woven cloth which is made from yarns of different materials (silk, cotton, nylon, etc.) interlaced in various patterns (e.g. plain, satin, twill). Fabrics present new challenges in differentiable modeling. First, the dynamics heterogeneity caused by material and structural diversity needs to be incorporated into modeling, which is especially crucial for solving inverse problems where the physical properties are learned from data. General DPMs without sufficient granularity can only approximate the dynamics and are unable to learn meaningful parameters. Second, certain forces that are essential for fabrics dynamics are indifferentiable. One such example is friction. The standard Coulomb model for rigid bodies has been made differentiable recently (de Avila Belbute-Peres et al., 2018b; Zhong et al., 2021). However, it is overly simplified for fabrics because the yarn-to-yarn friction shows richer dynamics (Zhou et al., 2017) that is beyond the capacity of existing methods. Further, the contact modeling together with friction requires new treatments that previous methods did not have to consider.

To overcome these challenges, we propose a new DPM for fabrics at a *more fine-grained* level and apply it to cloth modeling. Unlike general DPMs, we start with a fine-grained yarn-level model. By modeling each yarn individually, we provide the capacity of modeling fabrics with mixed yarns and different woven patterns, which could not be handled previously. To facilitate gradient-based learning, we propose new differentiable forces on/between yarns, including contact, friction and shear. Finally, we incorporate implicit Euler and implicit differentiation to compute gradients induced by an optimization problem embedded in the simulation.

To our best knowledge, our model is the first differentiable physics model which provides sufficient granularity for heterogeneous materials such as fabrics. We comprehensively evaluate its learning capability, data efficiency and fidelity. Since there is no similar model, we compare our model with the most similar work (Liang et al., 2019) and traditional Bayesian optimization on inverse problems. We also compare our work on control learning with popular Reinforcement Learning methods. We show that our model is more explicable, has higher data efficiency, generates more accurate predictions in inverse problem and control learning respectively.

## 2 RELATED WORK

**Differentiable physics simulator.** A differentiable simulator integrates differentiable physics engine into the forward and backward propagation of learning. As a strong inductive bias, these simulation engines increase data efficiency and learning accuracy over gradient-free models. Due to these advantages, differentiable simulation demonstrates superiority in a number of problem domains such as inverse problem, robot control and motion planning. The early works focused initially on simple rigid bodies (de Avila Belbute-Peres et al., 2018a; Degrave et al., 2019) and later simulation of high degrees of freedom systems, such as fluids (Schenck & Fox, 2018), elastic bodies (Hu et al., 2019; Huang et al., 2021), and cloth (Liang et al., 2019). More recently, Jatavallabhula et al. (2021) introduced an end-to-end differentiable simulator that can learn from images by combining differentiable rendering and differentiable simulation. Comparatively, we explore fine-grained DPMs for composite materials, which leads to new challenges in differentiable modeling.

**Cloth simulation.** Cloth simulation initially appeared in textile engineering and was then introduced to computer graphics (Long et al., 2011). Cloth has been modeled as particle systems (Breen et al., 1992), mass-spring systems (Provot et al., 1995), and continuum (Narain et al., 2012). Kaldor et al. (2008) proposed a yarn-level knit cloth simulator and found that cloth microstructures have a considerable influence on cloth dynamics. Since then the cloth simulation community has shifted the focus to yarn-level cloth simulation. Based on the objectives, the recent research can be classified to increasing efficiency (Kaldor et al., 2010; Cirio et al., 2016), combining continuum models and yarn-level models (Casafranca et al., 2020; Sperl et al., 2020), introducing woven cloth simulation (Cirio et al., 2014), and optimizations (Pizana et al., 2020; Sánchez-Banderas et al., 2020). Our work is orthogonal to these papers in that we introduce a new methodology to incorporate differentiable physics into yarn-level models.

**Machine learning and cloth simulation.** Machine learning was initially introduced to cloth simulation to make data-driven simulators, which have inherent advantages in simulation efficiency over physical-based methods (James & Fatahalian, 2003; Kim & Vendrovsky, 2008), and can help improve fidelity (Lahner et al., 2018). In parallel, machine learning has been applied to discover the physical properties from visual information. Bouman et al. (2013) proposed a linear regression model for evaluating cloth density and stiffness from the dynamics of wind-blown cloth. Yang et al. (2017b) introduced a neural network for classifying cloths based on how their dynamics are affected by stretching and bending stiffness. Rasheed et al. (2020) proposed a model for estimating the friction coefficient between cloth and other objects. By combining physically-based cloth simulators and neural networks, Runia et al. (2020) estimated cloth parameters by training neural networks to adjust a simulator's parameters so that the simulated cloth mimics the observed one in videos. Different from these gradient-free models, Liang et al. (2019) and Li et al. (2021) proposed sheet-level differentiable cloth models that can be used to estimate cloth parameters. In this work, we dive into fine-grained physics and propose a new yarn-level differentiable fabrics model which can be embedded into deep neural networks as a layer.

## 3 METHODOLOGY

Since cloth is employed as an application in this paper, we use the terms 'cloth' and 'fabric' interchangeably. We first explain the cloth representation (Sec. 3.1) and the (physics) system equation for simulation (Sec. 3.2). Then we present our new force models (Sec. 3.3), and how we solve the system equation to enable back-propagation (Sec. 3.4).

### 3.1 CLOTH REPRESENTATION

Similar to Cirio et al. (2014), our cloth consists of two perpendicular groups of parallel yarns named *warps* and *wefts*. Every pair of warp and weft are in contact with each other at one crossing node (Figure 1), with a persistent contact. We employ an Eulerian-on-Lagrangian discretization (Sueda et al., 2011), and denote the Degrees of Freedom (DoFs) of every crossing node as $\mathbf{q}_i \equiv (\mathbf{x}_i, u_i, v_i)$. $\mathbf{x}_i \in \mathbb{R}^3$ is the Lagrangian coordinates indicating spatial locations and $(u_i, v_i)$ is the Eulerian coordinates indicating sliding movements between yarns. The end points of yarns do not contact with other yarns and hence they are treated as special crossing nodes that have no Eulerian terms, i.e. $\mathbf{q}_j \equiv \mathbf{x}_i$. Therefore, on a $r(rows) \times c(columns)$ cloth, there are $(r-2) \times (c-2)$ crossing nodes with five DoFs and $2r + 2c - 4$ crossing nodes with three DoFs. Every two neighboring crossing nodes on the same warp/weft delimit a warp/weft segment. A warp segment with end points $\mathbf{q}_0$ and $\mathbf{q}_1$ is denoted as $[\mathbf{q}_0, \mathbf{q}_1]$ and its position is $(\mathbf{x}_0, \mathbf{x}_1, u_0, u_1)$ (Figure 1). This way, a woven cloth is discretized into crossing nodes and segments which are the primitive units of the cloth. Every segment is assumed to be straight so that linear interpolation can be employed on the segment, i.e. the spatial location of a point in the segment $[\mathbf{q}_0, \mathbf{q}_1]$ is $\mathbf{x}(u) = \frac{u-u_0}{\Delta u}\mathbf{x}_0 + \frac{u_1-u}{\Delta u}\mathbf{x}_1$, where $u$ is the point's position in Eulerian coordinates and $\Delta u = u_1 - u_0$ is the crossing nodes distance in Eulerian coordinates. We use $L$ to denote the rest length of the yarn segment and $R$ to denote the yarn radius.

### 3.2 SYSTEM EQUATION FOR SIMULATION

A cloth's state at time $t$, $\mathcal{S}_{(t)} = \{\mathcal{Q}_{(t)}, \dot{\mathcal{Q}}_{(t)}\}$, includes all the crossing node DoFs $\mathcal{Q} = \{\mathbf{q}_i | i = 1, 2, \ldots, N\}$ and their velocities $\dot{\mathcal{Q}} = \{\dot{\mathbf{q}}_i | i = 1, 2, \ldots, N\}$, where $N$ is the number of crossing nodes. Knowing the state, we can calculate the internal and external forces:

$$\mathbf{F} = \mathbf{M}\ddot{\mathbf{q}} = \frac{\partial T}{\partial \mathbf{q}} - \frac{\partial V}{\partial \mathbf{q}} - \dot{\mathbf{M}}\dot{\mathbf{q}} \tag{1}$$

where $\mathbf{q}$, $\dot{\mathbf{q}}$, and $\ddot{\mathbf{q}}$ are the *general* position, velocity, and acceleration respectively, with a dimension $l = 3 \times r \times c + 2 \times (r-2) \times (c-2)$. $\mathbf{M} \in \mathbb{R}^{l \times l}$ is the general mass matrix. The model assumes mass is distributed homogeneously. $T$ and $V$ are the kinetic and potential energy. As force is related to the partial derivative of energy with respect to position, the right hand terms in Eq. 1 are inertia, conservative forces, and part of the time derivative of $\mathbf{M}\dot{\mathbf{q}}$. Non-conservative forces are added to the right

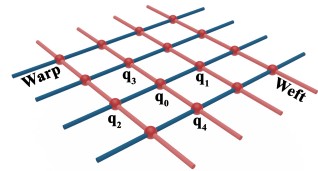

Figure 1: Blue and red rods denote warps and wefts respectively. **q**s are the crossing nodes.

side of the equation. Section 3.3 gives the details of all the forces considered in our model. Using implicit Euler (Baraff & Witkin, 1998), we can derive the system equation for simulation:

$$\left(\mathbf{M} - \frac{\partial \mathbf{F}_{(t)}}{\partial \mathbf{q}} h^2 - \frac{\partial \mathbf{F}_{(t)}}{\partial \dot{\mathbf{q}}} h\right) \dot{\mathbf{q}}_{(t+1)} = h \left(\mathbf{F}_{(t)} - \frac{\partial \mathbf{F}_{(t)}}{\partial \dot{\mathbf{q}}} \dot{\mathbf{q}}_{(t)}\right) + \mathbf{M}\dot{\mathbf{q}}_{(t)} \tag{2}$$

where the subscript in brackets $t$ indicates the associate variable at time $t$. Detailed deduction is in Appendix.

## 3.3 FORCE MODELS

To simulate cloth, we need to compute the inertia, internal and external forces in Equation 2. Inertia is the derivative of kinetic energy with respect to node positions. As woven cloths are interlaced yarns, the internal forces can be further classified into 1) forces caused by yarn deformation and 2) forces resulting from yarn-to-yarn interactions. We treat each yarn as an elastic rod that can generate elastic energy, including stretching and bending (Jawed et al., 2018), ignoring the twisting due to its triviality in cloth dynamics (Cirio et al., 2014). The elastic energy is $V^e = V^s + V^b$, where $V^s$ and $V^b$ are the stretching and bending energy respectively. The yarn-to-yarn interaction forces include friction, shear, and parallel yarn collisions. We refer the reader to Appendix for the inertia, stretching, and bending force as it is straightforward to show that they are differentiable.

**Yarn-to-yarn contact**. While the aforementioned forces are differentiable, the yarn-to-yarn forces are not. Existing differentiable contact models mainly correct after-contact positions and velocities (de Avila Belbute-Peres et al., 2018b; Liang et al., 2019; Zhong et al., 2021) via (multiple) optimization solves, which is too simplistic for fabrics. Yarn-to-yarn contact has its unique features. It is relatively sticky and often has small relative velocities. We need a contact model that reflects this and leads to a differentiable contact force which affects the friction/shear. The contact force is a combination of the stretching $\mathbf{F}_s$ and bending forces $\mathbf{F}_b$ at every crossing node along the contact normal $\mathbf{n}$. We assume no-slip contact and compute the contact force by:

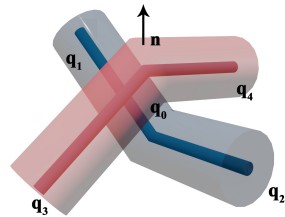

Figure 2: Compression force on $\mathbf{q}_0$ along normal $\mathbf{n}$ at $\mathbf{q}_0$.

$$F_n = \text{ReLU}(\frac{1}{2}\mathbf{n}^\top(\mathbf{F}_s^u + \mathbf{F}_b^u - \mathbf{F}_s^v - \mathbf{F}_b^v)) \tag{3}$$

where $u$ and $v$ represent the forces from warp and weft segments. The rectified linear unit (ReLU) ensures the non-negativity of the contact force. The normal $\mathbf{n}$, from warp to weft yarn (Fig. 2), is approximated by the normal of the best-fit plane of $\mathbf{q_0}$-$\mathbf{q_4}$.

**Friction**. The friction between warps and wefts prohibits relative movements, which is crucial to the overall dynamics of the fabric. In differentiable physics, contacts under simple settings have been modeled, such as the standard Coulomb model for kinetic friction (Zhong et al., 2021). But this is insufficient for our purpose for two reasons. First, the static friction plays a key role in stick-slip behaviours of yarns (Zhou et al., 2017) and needs to be modeled. Second, the standard Coulomb friction model is a piece-wise function, which is intrinsically indifferentiable at the static-to-kinetic transition point. Therefore, we need a new differentiable friction model.

The low relative speed between yarns is a special situation where the static-to-kinetic transition could actually be continuous (as opposed to the Coulomb model), experimentally shown by Stribeck (Stribeck, 1902). This indicates that a continuous and differentiable model has the potential to be more accurate for yarns than the widely used Coulomb model. Further, the breakaway force causing the static-to-kinetic transition depends on the rate of the external force (Johannes et al., 1973), and the nonlinear stick-slip behavior is related to self-excited vibrations before transition (Awrejcewicz, 1988). Inspired by the above research, we propose a new differentiable yarn-to-yarn friction model (Figure 3):

$$F_{Slide} = -\left(\frac{k_f \delta u - K(\delta u)\mu F_n}{2} K(\mu F_n - F_u) + \frac{k_f \delta u + K(\delta u)\mu F_n}{2}\right) - d_f \dot{u}_0 \tag{4}$$

where $\delta u = u_0 - \bar{u}_0$ and $K(x) = \tanh(px)$. $\bar{u}_0$ is the anchor position when there is no relative movement between the warp and the weft segment, $\mu$ is the friction coefficient, and $F_u$ is the external

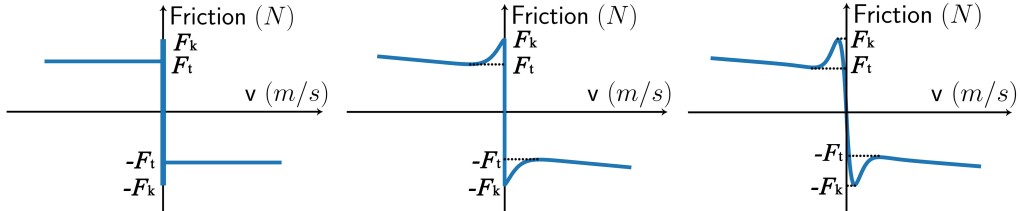

Figure 3: $F_k$ and $F_t$ are the static and kinetic friction. Coulomb model (left) is an indifferentiable multi-value function. Stribeck effect (middle) is empirically observed (Stribeck, 1902). Our model (right) incorporates the Stribeck effect and also simulates self-excited vibrations around $\mathbf{v} = 0$.

force. We introduce a hyperparameter $p$ to control the conversion speed between static and kinetic friction. To understand Equation 4, there are three situations: $F_u = 0$ (no external force), $0 \leq F_u \leq \mu F_n$ (static friction), and $F_u > \mu F_n$ (kinetic friction). When $F_u = 0$, $\delta u = 0$, $K(\mu F_n - F_u) = 1$ and the speed $\dot{u}_0 = 0$, so $F_{Slide} = 0$; when $0 \leq F_u \leq \mu F_n$, we allow a small displacement $\delta u$ to mimic the self-excited vibration in static friction, governed by a Hooke's spring $k_f \delta u$ with stiffness $k_f$. When $F_u$ is small, i.e. $K(\mu F_n - F_u)$ is close to 1, $F_{Slide} \approx -k_f \delta u - d_f \dot{u}_0$ where $d_f$ is a damping coefficient. $F_{Slide}$ is mainly the static friction minus a small damping term (as $\dot{u}_0$ is small). Due to the time discretization in simulation, the spring force has a delayed response which causes small-range vibrations. When $K(\mu F_n - F_u)$ starts to decrease to 0 and the breakaway force is achieved $F_u = \mu F_n$, $F_{Slide} = -\frac{k_f \delta u + K(\delta u)\mu F_n}{2} - d_f \dot{u}_0$ and $\frac{k_f \delta u + K(\delta u)\mu F_n}{2}$ is the average of the spring force and the maximum static friction. Finally when $F_u > \mu F_n$, $K(\mu F_n - F_u)$ quickly becomes -1 and $K(\delta u)$ becomes 1 as $\delta u$ increases. Then $F_{Slide} = -\mu F_n - d_f \dot{u}_0$, which is the kinetic friction minus damping. Fig. 3 shows our friction can closely approximate the Stribeck effect while maintaining differentiability, as opposed to the indifferentiable Coulomb model. Also, it incorporates self-excited vibrations within $F_{Slide} \in [-F_k, F_k]$ which is when $0 \leq F_u \leq \mu F_n$.

**Shear**. A shear force is generated when there is relative rotation between a warp and a weft at a crossing node (Parsons et al., 2010), which increases non-linearly when the shear angle increases (Mohammed et al., 2000; Peng et al., 2004; Cao et al., 2008). Previous differential models do not consider this type of forces. Therefore, we propose a new differentiable shear model. There are different stages when the shear angle increases (King et al., 2005). The shear force first grows almost linearly initially, then 'shear lock' is triggered (Wang

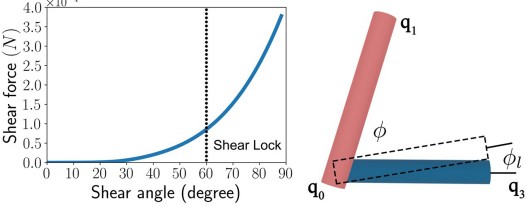

Figure 4: Shear force strength vs shear angle $\bar{\phi} - \phi$, (left) and graphical illustration (right).

et al., 1999) when the angle passes a threshold and the shear force starts to increase exponentially as the angle increases, producing highly non-linear behaviors. We therefore define the shear energy as a function of the shear angle $\bar{\phi} - \phi$ (Figure 4): $\frac{1}{2}k_s L(\phi - \bar{\phi})^2$, where $\bar{\phi} = \frac{\pi}{2}$ is the rest shear angle. $k_s = S\pi R^2(1 + F_n)$ is the shear stiffness and $S$ is the shear modulus. We embed the 'shear lock' by boosting $k_s$ exponentially with $\gamma = (\sqrt{2L^2 - 2\sin\frac{\phi}{2}L})/R$ as long as it stays smaller than the lock threshold $\phi_l = 2\arcsin\frac{R}{L}$: $k_s$ equals to $S\pi R^2(1 + F_n)$ if $\phi > \phi_l$; and $S\pi R^2(1 + F_n)\gamma^c$ otherwise, where $c$ controls the increase rate of $k_s$ with respect to $\phi$ when 'shear lock' occurs. Although $k_s$ has discontinuities within $[0, \frac{\phi}{2}]$, our new shear stiffness can be defined as:

$$k_s = \frac{1}{2}(F_n + 1)S\pi R^2\left((1 + \gamma^c) + (1 - \gamma^c)\tanh\left(\frac{\bar{\phi}^5(\phi - \phi_l)}{(\phi(\phi - \phi_l)(\phi - \bar{\phi}))^2 + \bar{\phi}^4\sigma^2}\right)\right) \quad (5)$$

where $\sigma$ governs the transition smoothness between lock and no-lock. The smaller the $\sigma$ is, the smoother the transition is. The shear force (Figure 4) derivation is in Appendix.

**Yarn-to-yarn collision**. The last internal force is the yarn-to-yarn collisions between parallel yarns. Although it is theoretically possible to use an existing approach (de Avila Belbute-Peres et al., 2018b;

Liang et al., 2019), it would require forming an optimization for all segments and therefore become prohibitively slow. Therefore, we introduce a new penalty energy, defined as a function of the nodes' distance in Eulerian coordinates for a warp segment $[\mathbf{q}_0, \mathbf{q}_1]$ (similar for a weft segment): $V_{0,1} = \frac{1}{2} k_c L (\text{ReLU}(d - \Delta u))^2$ where $d = 4R$ or $2R$ which is elaborated in Appendix.

**External forces and collisions**. Without loss of generality, we consider two external forces: gravity and wind force. Their impacts can be modeled by defining proper potential energies. Finally, after calculating all forces, the resultant force at every crossing node is the combined force of all segments that connect to that node. The cloth is simulated by solving Equation 2. We use bounding volume hierarchy (Tang et al., 2010) with continuous collision detection and non-rigid impact zones (Harmon et al., 2008) to compute collision. Similar to Liang et al. (2019); Wang et al. (2020), we form an optimization problem for continuous collision. Details can be found in Appendix.

### 3.4 DERIVATIVES OF THE SIMULATOR

Now we have a fully differentiable simulator with parameters $\boldsymbol{\omega}$. The $\boldsymbol{\omega}$ is the cloth physical parameters (stretching, bending, shearing, etc) when solving inverse problems, or the to be learned external forces in control experiments. Given a loss function $\mathcal{L}$, its gradient with respect to the parameters $\frac{\partial \mathcal{L}}{\partial \boldsymbol{\omega}}$ can help learn the right physics parameters via back-propagation. Implicit differentiation can be used to derive $\frac{\partial \mathcal{L}}{\partial \boldsymbol{\omega}}$ as detailed in Appendix. Finally, we define the loss function: $\mathcal{L}(\mathbf{q}, \hat{\mathbf{q}}) = \frac{1}{NT} \sum_{n=1}^{N} \sum_{t=1}^{T} \|\mathbf{q}_{n,t} - \hat{\mathbf{q}}_{n,t}\|_2^2$, where $\mathbf{q}_{n,t}$ and $\hat{\mathbf{q}}_{n,t}$ are the ground-truth and predicted general position. $N$ and $T$ are the total number of nodes and simulation steps respectively. We use Stochastic Gradient Descent and run 70 epochs for training.

**Underconstrainedness Mitigation.** Learning physical parameters via solving an inverse problem is intrinsically under-constrained, leading to multiple solutions or implausible parameter values when fitting data, e.g. unconstrained learning leads to negative density. We mitigate this issue by incorporating prior knowledge. Instead of directly learning parameters $\boldsymbol{\omega}$, we set $\boldsymbol{\omega} = a \times \text{sigmoid}(y) + b$ where $a$ and $b$ are tunable scalars, and we learn $y$ instead. $a$ and $b$ essentially limit the range of $\boldsymbol{\omega}$. We can induce prior knowledge of parameter ranges such as yarn density ranges, because although the exact value is to be learned and not known *a priori*, their ranges are available in practice. Our experiments demonstrate that this strategy effectively mitigates the multi-solution issue.

## 4 EXPERIMENTS

We employ a traditional indifferentiable yarn-level simulator (Cirio et al., 2014) to generate the ground-truth data, and build a dataset of fabrics with three types of yarns and three types of woven patterns. The yarns vary in density, elastic modulus, and bending modulus (Table 1). The three woven patterns include plain, twill and satin. We use hybrid fabrics made from two types of yarns, and exhaustively combine three yarns with three woven patterns (Figure 5) to generate 9 types of fabrics. We denote them as XXX-(X, X) where the prefix is the woven pattern and the numbers in the brackets are the yarns, e.g. Plain-(1, 2) means a plain pattern woven with Yarn1 and Yarn2. In our ground-truth data, we use a square piece of cloth hanging at its two corners and blown by wind with a constant magnitude. The simulation is conducted for 500 steps with $h = 0.001s$. Training details are in Appendix.

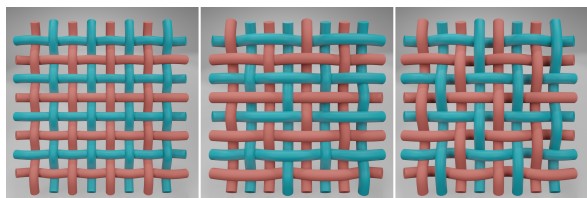

Figure 5: Woven patterns. Left-to-right: plain, satin, and twill. Teal and coral indicate different yarns.

### 4.1 LEARNING PHYSICAL PARAMETERS

We first demonstrate our model's effectiveness in learning meaningful physical parameters, under various model sizes and different amounts of training data.

**Learning capacity.** We first test whether meaningful physical parameters can be learned from cloths of different sizes. Small-size cloths tend to show low-frequency features, e.g. the general shape, as

Table 1: Ground-truth parameters of three yarns.

| Parameter/Yarn | Yarn1 | Yarn2 | Yarn3 |
|---|---|---|---|
| Density($kg/m$) | 0.0020 | 0.0025 | 0.0024 |
| Stretch modulus($N/m$) | 500000 | 170000 | 120000 |
| Bending modulus($N/m$) | 0.00014 | 0.00011 | 0.00009 |

opposed to high-frequency features, such as wrinkles and buckling. We test our model on simulation data with sizes: 5×5, 10×10, 17×17 and 25×25, trained on the first 25 frames. Table 2 shows that our model can effectively estimate yarn parameters with underlying physics models of different sizes. This has several implications. First, although cloth size does affect the overall dynamics of the motion in the ground-truth data (e.g. larger cloths have more wrinkles), it does not affect our model's learning capability. Second, since our model can reliably learn the yarn parameters on a small fraction of a cloth, it saves the computation of learning from large cloths, which improves the learning scalability. We can learn from small cloths and then scale to simulate large cloths. Further, inter-yarn parameters including shear $S$ and friction coefficient $\mu$ are highly correlated so that the model can easily end up learning only plausible parameter values rather than the true values. This is where we expect our model to suffer from the under-constrainedness problem as Liang et al. (2019). Surprisingly, our model can learn the right parameters under different sizes. By introducing prior knowledge as aforementioned, the learned parameters are restricted within valid ranges.

Table 2: Inter/intra parameters learned on Plain-(1, 2), with ground-truth $S = 1000Pa$, $\mu = 0.5$.

| Size | Shear $S$ | Friction $\mu$ | Yarn | Density | Stretch | Bend |
|---|---|---|---|---|---|---|
| 5 × 5 | 949 | 0.437 | 1 | $2.028 \times 10^{-3}$ | 479523 | $1.387 \times 10^{-4}$ |
| | | | 2 | $2.450 \times 10^{-3}$ | 172928 | $1.112 \times 10^{-4}$ |
| 10 × 10 | 932 | 0.455 | 1 | $1.991 \times 10^{-3}$ | 484719 | $1.325 \times 10^{-4}$ |
| | | | 2 | $2.448 \times 10^{-3}$ | 173843 | $1.026 \times 10^{-4}$ |
| 17 × 17 | 947 | 0.402 | 1 | $1.969 \times 10^{-3}$ | 505421 | $1.323 \times 10^{-4}$ |
| | | | 2 | $2.440 \times 10^{-3}$ | 171304 | $1.034 \times 10^{-4}$ |
| 25 × 25 | 913 | 0.380 | 1 | $2.069 \times 10^{-3}$ | 510215 | $1.488 \times 10^{-4}$ |
| | | | 2 | $2.443 \times 10^{-3}$ | 173920 | $1.201 \times 10^{-4}$ |

**Data efficiency.** Data efficiency is crucial as obtaining the ground-truth data can be expensive. Precise 3D geometry capture of real cloths is difficult and time-consuming, while simulation of high-res cloths is prohibitively slow. We further investigate the data efficiency by varying the amount of training data. We gradually increase the training data from the first 5 frames to the first 25 frames. Table 3 shows that our model has high data efficiency. It can learn reasonably well from as few as the first 5 frames. The benefits are two-fold. First, our model needs just a few frames to train, making it highly applicable. The second benefit is bigger but less obvious. The first 5 frames (from a static pose) normally contains little dynamics as the cloth just starts to move. This indicates that our model only requires a few frames of low-dynamics motions. This eases real-world measurements on cloth because no large motions are needed. This also saves time if simulation data is used, as small time step size is usually demanded in high-dynamic motion simulations.

Table 3: Plain-(1,2) learnt parameters on different training data. Left: Yarn1, Right:Yarn2.

| Frames | Density | Stretch | Bend | Density | Stretch | Bend |
|---|---|---|---|---|---|---|
| 5 | $2.030 \times 10^{-3}$ | 494301 | $1.357 \times 10^{-4}$ | $2.450 \times 10^{-3}$ | 169597 | $1.130 \times 10^{-4}$ |
| 10 | $2.037 \times 10^{-3}$ | 491717 | $1.379 \times 10^{-4}$ | $2.443 \times 10^{-3}$ | 169543 | $1.130 \times 10^{-4}$ |
| 25 | $2.038 \times 10^{-3}$ | 491873 | $1.367 \times 10^{-4}$ | $2.447 \times 10^{-3}$ | 167217 | $1.096 \times 10^{-4}$ |

All simulations can be found in the supplementary video. We also include simulations with collisions and simulations on large cloths using parameters learnt on small cloths. More results and details are in Appendix.

Table 4: Testing errors ($\times 10^{-6}$) of our model (left) and (Liang et al., 2019) (middle) and BO (right) trained on 5, 10 and 25 frames generated by yarn-level simulator (Cirio et al., 2014).

| Fabrics/Frames | 5 | 25 | 5 | 25 | 5 | 25 |
|---|---|---|---|---|---|---|
| Plain-(1,2) | $1.152 \times 10^{-4}$ | $3.962 \times 10^{-5}$ | 1.461 | 0.4124 | 0.512 | 0.109 |
| Plain-(1,3) | $1.516 \times 10^{-4}$ | $3.555 \times 10^{-5}$ | 1.608 | 0.4567 | 1.280 | 0.738 |
| Plain-(2,3) | $5.233 \times 10^{-4}$ | $2.117 \times 10^{-5}$ | 1.952 | 0.2294 | 28.19 | 18.16 |

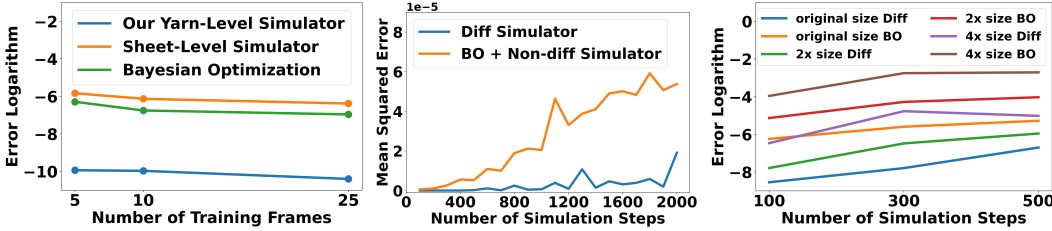

Figure 6: Simulation errors: data efficiency (left), long (middle) and big cloths simulation (right).

## 4.2 COMPARISONS

### 4.2.1 PREDICTION & DATA EFFICIENCY

To our best knowledge, there is no similar fine-grained DPM in the literature. The closest method is a general sheet model (Liang et al., 2019), so we compare our model with theirs. We employ their settings, and use a 17×17 model and 50 frames simulation data, with 5, 10 and 25 frames for training and the whole 50 for testing. Since the two methods model cloths at different levels of granularity, their physical parameters are not directly comparable. We therefore compare their Mean Squared Error (MSE). We also include a traditional parameter estimation method based on Bayesian Optimization (BO) (Snoek et al., 2012) combined with a yarn-level simulator (Cirio et al., 2014) as another baseline. In BO, we randomly select 5 initial points and use the expected improvement (Jones et al., 1998) as the acquisition function. As the learning process of differentiable simulation consists of forward simulation and backward simulation, training 70 epochs can be considered as running 140 simulations. Therefore, we run 140 iterations when using BO. Moreover, we impose the same parameter ranges in the BO as we did in our model.

From Table 4, our model uses data more efficiently than (Liang et al., 2019) and BO. From training on 5 frames to 25 frames, our model reduces the error by as much as 96% on Plain-(2, 3), while the largest improvements by the sheet-level model and BO optimization are 88% on Plain-(2, 3) and 78% on Plain-(1,2) respectively. Moreover, as shown in Figure 6 left, our error on 5 frames is already several magnitudes smaller than the baselines. Further reducing it requires the model to be able to learn subtle dynamics very well. Liang et al. (2019) essentially treats fabrics as a sheet. Since the simulation is from a yarn-level simulator (Cirio et al., 2014) which contains rich dynamics, the sheet model cannot precisely capture the subtle dynamics caused by individual yarns and their interactions. Further, the model granularity difference has more profound impact than just prediction. Being able to learn yarn parameters has immediate benefits for manufacturing and design, in terms of providing guidance on the choices of yarns and woven patterns. In addition, although BO can sometimes perform slightly better than the sheet model benefiting from a yarn-level simulator, its optimization process is not as efficient as ours. We only show results on Plain here and refer the readers to Appendix for Satin, Twill, and video comparisons.

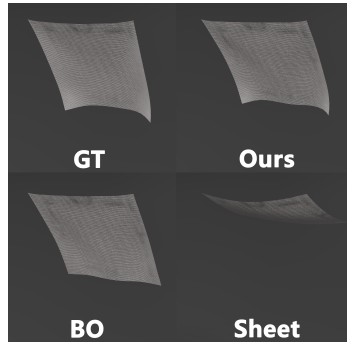

Figure 7: Simulation snapshots of the same step. The parameters estimated by our model is visually closest to the ground truth.

**Error significance.** The MSE errors in Table 4 seem to be small, this is because the cloth is small and only simulated for a short period of time. But the results suggest errors in parameter estimation, which are amplified when the cloth is larger and simulated for a longer time. To demonstrate this, first, we run forward simulations for 2000 steps with parameters learned by our model and BO.

Second, we show the compound influence using the parameters estimated by our model, Liang et al. (2019), and BO, and simulate a $17 \times 17$ cloth for 500 steps in the original size, 2 times size, and 4 times size. Fig. 6 middle-right show both results, which demonstrates the importance of accurate parameter estimation. The errors of BO and Liang et al. (2019) quickly become several times higher than our model when we scale the size and simulation time. We also show a visual comparison in Figure 7 and refer the readers to Appendix for more results.

### 4.2.2 CONTROL LEARNING

We also show that our model can facilitate control learning. We design a task with a cloth placed on a table and aims to learn forces applied onto the four corners of the cloth to throw it into a box next to the table. The forces are only applied in the first 5 frames. We use our model to learn a sequence of forces which can throw the cloth into the box and compare it with a reinforcement learning baseline model: PPO (Schulman et al., 2017). In addition, we also present a variant of our model by appending two fully-connected layers after our model output (Ours + FC). We use the center of the box's bottom as the target location. When training our model, we use the $l_2$ distance between the cloth center of mass and the target position as the loss. When training PPO, we use the same $l_2$ distance for the reward.

The result shows both our model and Ours+FC can quickly learn the forces to throw the cloth into the box. By contrast, PPO is model-free and much slower because it needs to sample in a huge action space. By contrast, the full differentiability of our model enables a quicker search for effective control forces. More details can be found in Appendix.

In a broader context, there are also model-free methods (Yang et al., 2017a; Pfaff et al., 2020) which can also learn physics. The differences between our model and theirs are: 1. model explicability. The parameters that our model learns are interpretable and have physical meanings, so that it can guide manufacturing and design. 2. data efficiency. The data efficiency is much higher in our method. Our model can use as few as 5 frames for learning while model-free methods typically require hundreds to thousands.

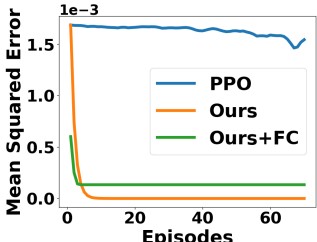

Figure 8: The MSE errors against epochs. Our approach learns faster than PPO.

## 5 DISCUSSION AND CONCLUSION

Our method is model-based, which requires domain knowledge and cannot simply 'plug and play' on data as model-free methods Pfaff et al. (2020). However, strong inductive biases from domain knowledge are necessary for differentiable physics to be applied in applications, because the model behaviour needs to be explainable in such applications, and cannot be merely black-box regression. Representative application domains include fabric manufacture/design and computer graphics, where both simulation and inverse problems need to be solved. Albeit focused on cloth, our model can be readily extended to general composite materials with mesh structures, e.g. from metal/plastic nets to buildings. In addition, our model can be embedded as a layer into a neural network, which helps learning control policies for cloth manipulation. Further, our model potentially enables a synergy between empirical physics modeling and deep learning, where our model can serve as a deterministic physics layer and other layers can incorporate non-linearity such as high-frequency dynamics in the system (Shen et al., 2021). Finally, our modeling of general forces such as friction and shear contributes to differentiable physical modeling in a wider range, given the universal presence of such forces in the real world.

To our best knowledge, we proposed the first yarn-level differentiable fabric simulator, in the pursuit of fine-grained DPMs capable of incorporating domain knowledge. Through comprehensive evaluation, our model can effectively solve inverse problems, provide high data efficiency and facilitate control. We investigated differentiable modeling of common forces such as friction and shear, which provides a foundation for future attempts on fine-grained differentiable physics modeling. In future, we will pursue other composite materials such as metal meshes. Also, we will explore more complex dynamics such as buckling and permanent damages.

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
