# OpenReview forum: "Fine-grained Differentiable Physics: A Yarn-level Model for Fabrics"
_ICLR.cc/2022/Conference — ICLR 2022 Poster_

### Official Review · Reviewer_Qcbf · 2021-10-24

**Correctness:** 3
**Technical Novelty And Significance:** 3
**Empirical Novelty And Significance:** 3
**Recommendation:** 6
**Confidence:** 4

**Main Review:**


Strengths:

This is the first fine-grained data-driven yarn-level model for describing internal forces for fabric. The differentiable nature of the forces allows this model to be incorporated in a learning-based cloth simulation framework, handling complex physical structures and heterogeneous materials at the yarn-level.   Experiment results show the model achieves significantly lower errors than previous methods and has high data efficiency.  Although only cloth simulation is demonstrated , the model can have wider applications in fabric manufacture and design areas.  The underlying principle may be extended more generally for other related tasks.


Weakness:

1. The paper incorporating some prior knowledge about the parameters’ range in order to mitigate illposedness.  However, the entire design process  is rather heuristic: neither analysis nor ablation study is presented to justify how such those design choices were made.

2. In order to illustrate data efficient of the proposed method,  the authors trained the same model on different frame numbers ranging from 5 to 25 for low-dynamics motions.   There is no mention of how the method performs as a function of the magnitude of the forces.  Additional experiments can be done and should be done.

3. In paragraph-2 of section 4.2.1, the paper claims “When the training data increase from 5 frames to 25 frames, the error reduces by around 100 times.  However, in Fig-6 left, the reduction of error is only 10-fold.  Similarly in Table-4,  the error is reduced only by about 10 times rather than 100 times, unless I misread this section.

4.  Experiments have been focused on simulated woven cloth.  Yet, the models are heavy-handedly designed. This casts some doubt regarding the generalizability of the proposed method. How does  the proposed force model on other different materials and weaving structures is unclear.


**Summary Of The Paper:**

This paper proposes a new differentiable physics-based model for simulating composite materials such as fabrics in cloths.  This is approached by formulating  several common differentiable forces exhibited in yarn-to-yarn interactions in different waving patterns.  Different from previous generally-purposed physics model, the proposed method is the first data-driven fine-grained yarn level modelling that takes into account complex structure/topologies and force interactions.  It models fabrics with composite materials and slim patterns and proposes new differentiable forces on/between yarns. Experiments show the efficacy of the proposed model, confirming the model’s capability for simulating heterogeneous materials such as fabrics.



**Summary Of The Review:**

Despite  that the experiment section of the paper can be improved in many aspects, I think the core contribution of the work is new and valuable, and the paper is acceptable condition on some minor revision.

---

> ### Author Response · Authors · 2021-11-18
> **Rationale Behind Parameter Prior, Changing Forces Magnitude, Errors in Writing, and Generalizability.**
>
> Thanks for your comments.
> # Ablation Study and Rationale Behind the Design of Prior Knowledge of Constraints
> We assume the reviewer refers to an ablation study on the parameter range constraints. We actually did an ablation study with/without such constraints at the early stage of this research. As we explained in the paper, the one without such constraints can often learn implausible parameters such as negative density, and consequently crash the system.  Further, adding such constraints allows the users to specify the general material properties if it is known (e.g., nylon or cotton).
> # Changing Force Magnitude
> Since the wind force is the major external force in our experiment, we assume that the reviewer referred to an experiment on the performance as a function of the magnitude of the wind force. Therefore, we also conducted an experiment using 5N (used in our paper), 10 N, and 15N wind force to blow a piece of 17x17 Plain-(1,2) cloth. The learning results are shown in the table below.
>
> |Wind Strength| Shear | $\mu$ | Density-1 | Stretch-1 | Bend-1 |Density-2 | Stretch-2 | Bend-2 |
> |:-:|-|-|-|-|-|-|-|-|
> |5|947|0.402|1.969e-3 |505421|1.323e-4 |2.440e-3 |171304|1.034e-4|
> |10|942|0.520|2.026e-3|494109|1.311e-4|2.441e-3|168267|1.049e-4|
> |15|934|0.586|2.029e-3|487918|1.341e-4|2.437e-3|167601|1.066e-4|
>
> The results show that our model can constantly accurately learn cloth parameters under different force magnitudes.
>
> # Errors in Writing
> Thank you for pointing out the error. You are right and we will rectify it as follows:  From Table 4, our model uses data more efficiently. From training on 5 frames to 25 frames, our model reduces the error by as much as 96% on Plain-(2, 3), while the largest improvements by the sheet-level model and BO optimization are 88% on Plain-(2, 3) and 78% on Plain-(1,2) respectively. Moreover, our error on 5 frames is already several magnitudes smaller than the baselines. Further reducing it requires the model to be able to learn subtle dynamics very well.
>
> # Generalizability
> Admittedly, there is a 'heavy-handledness' in our model which originates from the strong inductive bias introduced from fabrics physics. However, we argue that these bias is crucial for fine-grained differentiable physics models, as a natural way to achieve explainablity is through the first-principles approach. Further, while factors like the woven structure is specific to fabrics, the proposed force models are indeed general by themselves. They are based on physical phenomena that have been widely observed in the real world. One example is the Strebeck effect between two objects in contact with small relative speed, which is not limited to two yarns in fabrics. Naturally, the proposed simulator is not coupled with the materials or woven structures in fabrics. As an example, we showed an experiment in Appendix A.1 where we fit a fabric to a mesh data where the woven structure is unknown. As a data fitting practice, it still performs well. Finally, as a feasibility study, this research focused on one type of composite material: fabric. It is indeed in our plan to test the model on other composite materials such as wire mesh. We will add this in discussion.

---

> > ### Comment · Reviewer_Qcbf · 2021-11-29
> > **confirmed**
> >
> > Dear authors, thank you for your careful responses. I believe the paper is acceptable.

---

### Official Review · Reviewer_e46u · 2021-10-31

**Correctness:** 4
**Technical Novelty And Significance:** 2
**Empirical Novelty And Significance:** 2
**Recommendation:** 6
**Confidence:** 3

**Main Review:**

Strengths:
- The proposed model allows to estimate physical parameters efficiently, ie. :
    - from a small sized fabric only (then simulations can be carried on a larger cloth size) (Table 2)
    - from a few frames only (table 3)
- The comparison of (a) gradient descent on the proposed model with (b) bayesian optimisation and (c) a sheet model turns favorable for (a) in terms of error + convergence speed.
- The force model is extensive (inertia, stretching, bending, contacts between yarns, friction, shear, collision between parallel yarns, gravity, wind…)
- The submission is well grounded in previous literature for yarn modelling
- Supplementary videos are a plus
- Nice figures and good readability

Weaknesses:
- In 4.1 the “Learning” experiments should rather be called “fitting” a few parameters of the model. Indeed the applicability is very narrow: the prior on parameters is strong (restricted range), the model is only able to fit data for which there is a solution (it is a simulated yarn fabric) and for which we have the exact positions of the intersections and the number of yarns. This setting is unrealistic. What would happen if we were to fit a 3D mesh, or a 3D sparse scan, for which the positions q_i are not available?
- An ablation study is missing: what would happen if the proposed differentiable forces were removed? Showing such a result would prove the relevance of devising a differentiable approximation for them
- Fitted parameters $\omega$ should be re-stated at the beginning of 3.4
- The scalability claim does not really hold: the largest demonstration (support video) is for a 70*70 yarns which is still very crude.


Minor typos:
- End of introduction “… and fast in control learning”: it looks like a word is missing
- Related work / differentiable physics simulator “simulaiton” -> simulation
- Beginning of Methodology (+repeated): “system equation” -> system of equations?
- In 3.2 the dimension $l=3rx+2(r-2)(c-2)$ is not equal to what is stated in 3.1 : $(r-2)(c-2)$ crossing nodes with 5 DoFs and $2r+2c$ with 3 DoFs gives a dimension of $l’=l+12$


**Summary Of The Paper:**

This submission proposes a differentiable parametric force model for fabric, at the level of yarns, by modelling interactions forces between yarns. Since some of these forces are not differentiable, the authors propose a differentiable approximation for them.
As a results, gradient descent can be used to recover physical parameters, by fitting the parametric model to a dynamic sequence of frames. Differentiability is also used to optimize forces for control: gradient descent can optimize the required forces to apply to a piece of fabric to throw it in the air and aim for a target position.


**Summary Of The Review:**

This paper proposes a new differentiable force model for yarn fabrics which is novel and able to fit clean data for which the positions of the yarns are known. Yet the experiments fail to prove it is useful to fit general fabrics in a less controlled setting.

---

> ### Author Response · Authors · 2021-11-18
> **Narrow Applicability, Removing Differentiable Forces, Restate Parameters, and Scalability.**
>
> Thanks for your comments.
> # Narrow Applicability
> Imposing ranges on parameters might not seem ideal. However, employing prior knowledge is a necessary step and standard practice in system identification nowadays (Hahn et al., 2019).  Further, the ranges might appear 'restricted' as they look small. However, small parameter changes in the parameter space correspond to significantly different materials. In this sense, the parameters are merely constrained within a range of one material type, not to a specific material. Also, we showed an experiment in the appendix where we fit our model to a 3D mesh, where we do not know the q_i positions or the ground-truth parameter values. The q_i positions were obtained by establishing a correspondence between the mesh vertices and the crossing nodes in our model. Our model performs well in fitting the dynamics. The 3D sparse scan is a wonderful idea, which will certainly enhance the applicability of our model. We will discuss it in the paper.
> # Removing Differentiable Forces
> Since the fabric would not be able to hold itself together if the forces are removed, we assume the reviewer meant what would happen if their indifferentiable counterparts are used. If we use the original indifferentiable force models, the simulator will be indifferentiable and not able to do gradient-based learning. Consequently, methods that do not make use of the gradient information like Bayesian Optimization (BO) will need to be used for estimating the physical parameters. We have shown comparisons with BO to prove the superiority of our method. In this sense, BO can be seen as an ablation study on solving the inverse problem.
> # Restate Parameters
> We will revise the paper accordingly.
> # Scalability
> To our differentiable cloth simulation, scalability includes two aspects: learning scalability and simulation scalability. The learning scalability refers to the model's capability of learning from different cloth sizes and numbers of frames. We demonstrated it in Table 2 and Table 3, where we also showed that it is possible to learn the physical parameters on a small cloth in a small number of frames, so that the learning scalability is not an issue. In addition, the simulation scalability refers to using the learned parameters to simulate large cloths. Since our main contribution is to replace discontinuous forces with continuous ones without adding much extra complexity to the physics model itself, the time complexity of our model is similar to the yarn-level simulator (Cirio et al., 2014). Indeed, our current implementation is CPU-based in C++. More engineering work will be put into a GPU implementation for larger demos. We will add discussions regarding this.
> # Minor Typos
> We will fix the language problems.

---

### Official Review · Reviewer_8QNo · 2021-11-03

**Correctness:** 3
**Technical Novelty And Significance:** 4
**Empirical Novelty And Significance:** 3
**Recommendation:** 6
**Confidence:** 4

**Main Review:**

Strengths:
* The paper studies a challenging topic in differentiable physics, namely, the fine-grained differentiable cloth simulation.
* The paper contains detailed problem formulation and derivations.
* The paper demonstrates promising results on the synthetic data compared to the baselines [Liang et al., In NeurIPS 2019] and Bayesian optimization on material estimation.

Weaknesses:
* The paper does not seem to comment on the computational cost and comparisons to baseline methods. It would be good to provide such information in the main paper, especially when compared with non-differentiable simulator [Cirio et al., 2014] and prior work on differentiable cloth simulation [Liang et al., In NeurIPS 2019].
* The quantitative results can be presented in a clearer manner. For example, it would be good to show the material prediction error in percentage in Table 2-4 (rather than the absolute value).  It would be good to include the algorithm (refer to Section 3.1 in [Liang et al. 2019]) in the main paper.
* It would be good to know whether the proposed differentiable simulation is deterministic or not? Whether the material estimation is sensitive to initialization? In this case, please consider reporting the standard deviation for multiple runs of the same setting.
* It is unclear what happens if the parameter prior was not provided to the proposed method in the control learning experiment. It would be good to provide such ablation studies and discuss the results in the rebuttal.

**Summary Of The Paper:**

This paper studies the fine-grained cloth simulation using differentiable physics models. The paper is largely based on the prior work on cloth simulation [Liang et al., In NeurIPS 2019] but with the domain-specific yarn-level formulation. Specifically, it proposes to represent cloth as two perpendicular groups of parallel yarns. Built upon this representation, the paper introduces several differentiable strategies to model inertia, internal and external forces. Experiments have been conducted on simulated data for material estimation and model-based control learning.

**Summary Of The Review:**

Overall, this is an interesting paper with solid technical contributions. The main weaknesses include the presentation and experimental evaluations.

---

> ### Author Response · Authors · 2021-11-18
> **Computational Cost, Result Showing, Random Initialization, Prior Constraints.**
>
> Thanks for your comments.
> # Not Largely Based on (Liang et al, 2019)
> There might be a misunderstanding. Our research is not largely based on (Liang et al., 2019), which is a sheet-level cloth model. Although we draw inspiration from it especially in collision detection/response, our model is rooted at a radically different level of granularity to capture the yarn-level dynamics.
> # Computational Cost
> The code of the non-differentiable simulation (Cirio et al., 2014) has not been released. We implemented a CPU version of this simulator following the paper, but the original simulator runs on CPU+GPU. Therefore, it would be unfair to do a direct performance comparison. Although a direct numerical comparison is difficult, it is easy to see the time complexity of these two simulators are similar. Our differentiable forces only incur small extra computation.
>
> Here, we also give performance comparison between our model and (Liang et al., 2019). On 17 x 17 square cloth, the average forward simulation time of (Liang et al. 2019) is approximately 0.7sec/step. Its average time in computing gradient is 1.5sec/step. Our average forward simulation time is 1.5sec/step and our average time in gradient computation is about 13sec/step. Considering that our simulator is more fine-grained, e.g. more internal forces, our forward simulation is not too much slower.
> # Result Presentation and Algorithm
> We calculate the prediction error percentages against the ground truth in [**Table 2**](https://drive.google.com/file/d/1EO1b43fc9hGOMGEjjNM5cvnNSBWjOhEs/view?usp=sharing) and [**Table 3**](https://drive.google.com/file/d/1sZQfS6QSfoBVDvm6dVKxvFhubmoSpVKA/view?usp=sharing). For Table 4, it is however not immediately clear what percentages the reviewer was referring to. As it is the MSE, so the ground truth MSE is zero, where direct error percentage calculation would involve dividing the error by zero. Here we assume the reviewer refers to the error percentage of the prediction against the initial error, similar to the error percentages in (Li et al, 2021), and show them in [**Table 4**](https://drive.google.com/file/d/1TQo-bmNaF24wQZwd7kn_WCtV9X-WyAJF/view?usp=sharing). We can recompute them if the reviewer can specify otherwise. In addition, we also provide the [**Algorithm**](https://drive.google.com/file/d/1CKblJNlnizGm48akL1BSxOGmn19NpWst/view?usp=sharing) of our simulator.
>
> # Deterministic or Not?
> Our differentiable simulator is deterministic. The material estimation results are affected by initialization but not sensitive. We report the mean and the standard deviation of multiple experiments. The initial values of the physical parameters are randomly selected from a range of ±10％ of the average of the two yarns. For instance, in learning the stretch in Plain-(1,2), we only know the ranges of the stretching parameters Y1 and Y2 of Yarn1 and Yarn2 but not the exact values. Therefore, when initializing Y1 and Y2, we randomly sample values from a range of ±10％ of the mean stretch stiffness of the Yarn1 and Yarn2, [mean(Y1, Y2) * 0.9, mean(Y1, Y2) * 1.1] for initialization. This is to test whether our model can stably learn the parameters. The results of the 5 repetitions are shown below, which are not very different from the result reported in the paper. Given that the standard deviations are small, it shows that our model can stably learn reasonable parameter values.
>
> |Size|Shear|$\mu$|Density-1|Stretch-1|Bend-1|Density-2|Stretch-2|Bend-2|
> |-|-|-|-|-|-|-|-|-|
> |$5\times5$|1011.79±6.12|0.39±0.08|1.98e-3±3.00e-5|498595±8862|1.37e-4±1.41e-6|2.45e-3±4.81e-5|186710±3776|1.11e-4±4.78e-6|
> |$10\times10$|983.41±6.84 |0.44±0.03|2.03e-3±5.04e-5|542375±7099|1.44e-4±2.08e-6|2.47e-3±4.73e-5 |180032±1848|1.05e-4±8.18e-6|
> |$17\times17$|962.29±8.99 |0.47±0.06|2.00e-3±6.66e-5|519993±3175|1.43e-4±5.55e-6|2.47e-3±5.04e-5|176232±1514|1.19e-4±6.50e-6|
>
> # No Prior Constraints in Control Experiment
> In the control learning experiment, cloth physical parameters are not trained. Therefore, we did not constrain these parameters in this experiment. Instead, the trained parameters are the external forces applied onto the four corners of the cloth. We do not add prior constraints on these forces, either. We followed the widely adopted setting in existing differentiable physics research (Du et al., 2021; Li et al., 2021; Liang et al., 2019) to design our control experiment.
>
> ## Additionaly References
> Du, T., Wu, K., Ma, P., Wah, S., Spielberg, A., Rus, D. and Matusik, W., 2021. DiffPD: Differentiable Projective Dynamics with Contact. *arXiv preprint arXiv:2101.05917*.
>
> Li, Y., Du, T., Wu, K., Xu, J. and Matusik, W., 2021. DiffCloth: Differentiable Cloth Simulation with Dry Frictional Contact. *arXiv preprint arXiv:2106.05306*.

---

### Official Review · Reviewer_u4Fx · 2021-11-03

**Correctness:** 4
**Technical Novelty And Significance:** 3
**Empirical Novelty And Significance:** 3
**Recommendation:** 6
**Confidence:** 2

**Main Review:**

Pluses:
* The authors identify a major shortcoming of previous simulation models, namely the use of discontinuous force terms, and engineer alternative models that approximate the terms from domain literature but have the added benefit of being smooth everywhere. I have to say I quite like this general approach of searching for the sweet spot between textbook physics and usability of the model for solving secondary problems.

Questions:
* You write that the non-linear stick-slip force term lends itself to oscillatory behavior. Without being familiar with such simulations in the slightest, I feel that this could lead to free energy being trapped in entropy and therefore thermal loss. Did you ever encounter such behavior?

* Why do differences between inter/intra parameters (Table 2) and ground truth as the grid size increases? Should it not be the other way around?

Weaknesses:
* In the Comparison video, I could not spot any significant difference between the different weaves. At least not more than between Ground Truth and Ours. Did you also experiment with a bit more extreme differences in fabric?

* It seems that no real-world experiments were done. The range of simulated experiments is not very wide, so I can't judge the expressivity / gamut of the model.

**Summary Of The Paper:**

This paper introduces a model for yarn-based fabric simulation that is differentiable and hence particularly well suited for solving inverse problems, such as deriving model parameters from reference video input.

The technical contribution itself (execution of the forward model) does not necessarily fall into the realm of machine learning but it is nicely embedded in an inverse framework and demonstrated on a scenario that is clearly computer vision. Hence, I would consider the paper on-topic for the venue.



**Summary Of The Review:**

Again, I am no expert, but I think this is a paper that fits its time. It appears to be well executed and evaluated. Whether or not the works falls under "machine learning" is probably a matter of personal taste. The contribution is a simulation that serves as forward model component in a regression framework, so I think that should qualify it.

As the authors acknowledge, neither the specific problem nor the general idea of differentiable simulation is new. Whether the specific solutions introduced in the paper generalize to other problems is unclear. Still, I think this is reasonably nice work to warrant acceptance.

---

> ### Author Response · Authors · 2021-11-18
> **Thermal Loss, Error Increase with Size, Woven Pattern, and Real-World Experiments**
>
> Thanks for your comments.
> # Energy Trapping and Thermal Loss
> The question on free energy is very insightful. When designing the friction force model, if the oscillation is not controlled, there is a danger of free energy being trapped.Therefore, we added a damping term in Equation 4 to avoid uncontrollable oscillations. Although our system does not explicitly consider thermal dynamics, the damping term is a straightforward way to ensure no energy trapping, which tends to stop oscillations over time.This question actually leads to a broader modeling question down the direction of the research.Since thermal dynamics can affect the physical properties of certain materials, we will add it in the discussion.Thanks for the question!
> # Parameter Estimation Error Increase with Size
> The observation in Table 2 regarding the grid size vs learning accuracy is likely to be caused by small randomness in the learning. This randomness results from that there are multiple sets of parameters with similar values which can all lead to the same simulation result. Fundamentally, this is because forces can compensate each other to achieve the same dynamics.
>
> To show the learning robustness, we report the mean and the standard deviation of multiple experiments with random initial parameters. In detail, the initial values of the physical parameters are randomly selected from a range of ±10％ of the average of the two yarns. For instance, in learning the stretch in Plain-(1,2), we only know the ranges of the stretching parameters Y1 and Y2 of Yarn1 and Yarn2 but not the exact values. Therefore, when initializing Y1 and Y2, we randomly sample values from a range of ±10％ of the mean stretch stiffness of the Yarn1 and Yarn2, [mean(Y1, Y2) * 0.9, mean(Y1, Y2) * 1.1] for initialization. This is to test whether our model can stably learn the parameters. The results of the 5 repetitions are shown below. Which are note very different from the result reported in the paper. Given that the standard deviations are small, it shows that our model can stably learn reasonable parameter values.
>
> |Size|Shear|$\mu$|Density-1|Stretch-1|Bend-1|Density-2|Stretch-2|Bend-2|
> |-|-|-|-|-|-|-|-|-|
> |$5\times5$|1011.79±6.12|0.39±0.08|1.98e-3±3.00e-5|498595±8862|1.37e-4±1.41e-6|2.45e-3±4.81e-5|186710±3776|1.11e-4±4.78e-6|
> |$10\times10$|983.41±6.84 |0.44±0.03|2.03e-3±5.04e-5|542375±7099|1.44e-4±2.08e-6|2.47e-3±4.73e-5 |180032±1848|1.05e-4±8.18e-6|
> |$17\times17$|962.29±8.99 |0.47±0.06|2.00e-3±6.66e-5|519993±3175|1.43e-4±5.55e-6|2.47e-3±5.04e-5|176232±1514|1.19e-4±6.50e-6|
>
> # Visual Difference Caused by Woven Patterns
> The woven patterns employed in the paper are not created by ourselves but from standard templates in textile. We do agree that visually they look similar under the current experiment setting. However, the woven patterns do affect the dynamics. To show this, we conducted simulations with the same parameters, but under different initial states. We shear three pieces of cloth in the initial state and then release them. They only differ in woven patterns.  For clarity, we turn off gravity and wind. The [**Figure**](https://drive.google.com/file/d/1j7SN_o0LKlsdq6c_fRj5FqHLdEOEzOGX/view?usp=sharing) shows top view of three pieces of cloth with different woven patterns in the initial step and 10 steps later.As shown in the figure, there are obvious differences after merely 10 steps. This demonstrates woven patterns have considerable influences on the overall mechanical properties.
> # No Real-World Experiments
> We do agree that having real-world experiments is desirable and it is exactly what we are doing in a bigger research plan. However, we argue that the first step on this line of research is to test the feasibility of differentiable yarn-level simulator, which is exactly what this paper is about.  Currently, we are obtaining real-world data, not only at the macroscopic sheet level but also at the yarn level with the mechanical properties measured by specialized equipment. This will bring about a new dataset for fine-grained differentiable physics research. We will add it into the future work.
> # Neither Cloth Simulation nor Differentiable Physics is New
> Although cloth simulation is not new, differentiable physics is a very new field. Within it, the first-principles approach to fine-grained differentiable physics is very new. To our best knowledge, this is the first time differentiable cloth modeling has been extended to the yarn-level. Through experiments, we demonstrate that our model can learn yarns’ physical parameters due to its finer granularity, which can potentially guide manufacturing in the way that no other methods can. Compared with existing sheet-level models, these parameters are more explainable.

---

> > ### Comment · Reviewer_u4Fx · 2021-11-29
> > **Thank you**
> >
> > Dear Authors, thank you for your very helpful response. The new video convinces me; please include it in the final version (in case of acceptance)

---

### Official Review · Reviewer_U3Yi · 2021-11-03

**Correctness:** 3
**Technical Novelty And Significance:** 3
**Empirical Novelty And Significance:** 4
**Recommendation:** 8
**Confidence:** 4

**Main Review:**

I believe the implementation of this paper is respectful and non-trivial. It looks like that this paper fills the gap in differentiable simulation with a cloth simulator from a yarn-level perspective. Despite its potential applications, I have a few technical questions and suggestions about it:

1. This manuscript provides a novel view of the intersection of differentiable simulation and cloth simulation by thinking at the yarn level. I believe one of the important baselines is [1], which was the latest work working on differentiable cloth simulation in a mesh-based view. I expect some discussion between this work and [1].
2. The statement "Previous research (Liang et al., 2019; Du et al., 2021) suffers from this problem, which unfortunately leads to learning implausible parameter values" looks technically inaccurate to me. Actually, leveraging prior knowledge for physical parameters is now a necessary step for system identification tasks. For example, because of the huge numerical difference between Young's modulus and Poisson's ratio, [2] proposes a logarithm transform to alleviate the unbalancement.
3. I am curious about the wall-clock time and memory footprint of the experiments. It will be helpful to note them.
4. As I questioned in the summary: a physical simulator usually comes with a steep learning curve which disables following works by the implementational difficulty. I wonder if the source code will be released for future follow-up.
5. My main concern is motivation. The yarn-level simulation was proposed for more accurate collision handling and detailed presentation. However, the experiments are too coarse to show these advantages, which makes me question why the yarn-level simulator is used. As a reference, [3] used much more yarns to represent their cloth. I expect a more valid explanation for motivation or a set of better results that are more complex than 25*25.

[1] Li, Yifei, et al. "DiffCloth: Differentiable Cloth Simulation with Dry Frictional Contact." arXiv preprint arXiv:2106.05306 (2021).

[2] Hahn, David, et al. "Real2sim: Visco-elastic parameter estimation from dynamic motion." ACM Transactions on Graphics (TOG) 38.6 (2019): 1-13.

[3] Sperl, Georg, Rahul Narain, and Chris Wojtan. "Homogenized yarn-level cloth." ACM Trans. Graph. 39.4 (2020): 48.

**Summary Of The Paper:**

This manuscript enhances the yarn-level cloth simulation to a differentiable level by introducing alternative differentiable operators in place of originally discontinuous/indifferentiable counterparts. In addition to differentiable components for energies, collisions, frictions, etc, this manuscript also differentiates through the implicit Euler method, which enables greater time steps with better stability. The authors empirically studied the efficacy of this differentiable simulator on a variety of environments with multiple baselines. I believe one of the major contributions can be a non-trivial implementation in C++ and its interfaces with modern deep learning architectures (unsure if it will be open-sourced through).

**Summary Of The Review:**

This manuscript proposes a combination of yarn-level cloth simulation and differentiable simulation and shows the application of system identification and control. The thing I am worried about is that the motivation of using yarn-level, which is originally designed for fine details but applied on over-simplified models. I would like to know more about why the authors chose the yarn-based method, how they would like to use this simulator, and where they see the simulator fits in. Generally, I appreciate the work but was not convinced by the motivation. I am open to changing my score based on the clarifications I get.

---

> ### Author Response · Authors · 2021-11-18
> **Compare with DiffCloth, prior knowledge, wall clock time, sharing code, and motivation.**
>
> Thanks for your comments.
> # Compare with [1]
> The primary difference is they are designed at different granularities. While[1] is a sheet-level model,ours is a yarn-level one. Consequently, our model can learn explainable and fine-grained physical parameters such as inter/intra yarn properties. This opens up potential applications not only for computer graphics but also manufacturing guidance and textile design. In addition, while[1] mainly deals with external contact forces, ours focuses on the internal yarn-to-yarn forces.So our research is orthogonal to [1]. Finally, our differentiability strategies are different. [1] accepts minor discontinuity in contact and shows backpropagation can still be done if they are rare,while ignoring large discontinuity which causes bumpy loss landscape (Fig.3 in[1]).By contrast,our strategy is to approximate the discontinuous forces with continuous ones based on empirical observations (e.g. the Strebeck effect) and ensure the full smoothness of the loss landscape,shown here [**landscapes**](https://drive.google.com/file/d/1mFSlh0BwzgVfLJtJVguq48CFXStaZEay/view?usp=sharing). We will add the discussion in the paper.
> # Prior Constraint
> Our statement on the previous work learning implausible parameters comes from our experiments using the code from (Liang et al., 2019) in our experiments. In that simulator, the prior knowledge is introduced in the initial guess, but we found it was not sufficient to stop it from learning implausible parameters such as negative density in our experiments. We will clarify it. In addition, thanks for the reference [2].The logarithm transformation is a neat idea but it does not provide a way to constrain the range. In our case, a value range normally corresponds to one type of materials (e.g. nylon or cotton) where prior knowledge is available. In this sense, our strategy is orthogonal to [2] in that ours brings a way to introduce stronger prior knowledge if available. We will add more discussions.
> # Wall Clock Time and Memory Footprint
> Average **simulation wall clock time** of cloths in different sizes:
>
> |5x5|10x10|17x17|25x25|
> |-|-|-|-|
> |95ms/step|465ms/step|1302 ms/step|3400ms/step|
>
> Approximate **simulation memory consumption** of cloths in different sizes:
>
> |5x5|10x10|17x17|25x25|
> |-|-|-|-|
> |40 MB|100MB|450MB|1.9G|
>
> The **learning wall clock time with 25 frames** of cloths in different sizes :
>
> |5x5|10x10|17x17|25x25|
> |-|-|-|-|
> |13sec|106sec|328sec|1310sec|
>
> The **learning wall clock time of 17x17  cloth** with different number of frames :
>
> |5steps|10steps|25steps|
> |-|-|-|
> |68sec|133sec|328sec|
>
> Note that our implementation is a naïve CPU-based one, without optimization,mainly aiming for proving the feasibility. There are mature techniques to migrate this to GPU for faster execution.We will add this into the future work.
> # Sharing Code
> Code will be shared upon acceptance.
> # Motivation
> We did show a 70x70 example in the video,but we understand the reviewer’s point [(**see the video**)](https://youtu.be/n5mfHe0Lmo4).
>  Admittedly, yarn-level simulation is good for finer collision handling and detailed presentation in computer graphics,but accurate physical parameter estimation is needed in other fields such as textile, which is beyond good visual effects. This is especially so when it comes to guiding textile design/manufacturing. Our model is, albeit applicable to, not only designed for visual effects. In this sense, it is complementary to research like [3]: our model can be used to learn the parameters then the learned yarn-to-yarn forces can then be incorporated in simulators like [3]. So our yarn-level modeling is highly desirable and can work with other yarn-level simulators. By contrast, existing sheet-level simulators (regardless of differentiability) cannot learn at this granularity. Furthermore, as a research topic, scientific curiosity is another strong motivation for us. Existing research tends to get deeper and finer levels, e.g. the development in thermodynamics when physics moved from bulk descriptions of gases to models based on the actual motions of large numbers of particles. As a combination of machine learning and physics, we strongly believe differentiable physics is one promising and natural direction to explore finer and finer granularity. Finally, in terms of the scalability of the learning, we have demonstrated that it is possible to learn the parameters on a small cloth in a small number of frames,then use them to run larger cloths.When scaling up, our computational complexity is similar to(Cirio et al. 2014), where the major limiting factor is the number of nodes. As we explained before, a GPU implementation and large-scale experiments with more complex topologies will be included in future work.

---

> > ### Comment · Reviewer_U3Yi · 2021-11-29
> > **Thank you for the responses**
> >
> > Dear authors,
> >
> > Thank you for your responses. I am convinced about the motivation, which is my major concern, and agree with the authors that it opens up a gate if the generalizability is reasonable. It looks like an important piece in the puzzle of differentiable simulation. I would like to raise my score to 8. Including the discussion in the manuscript (if accepted) would be my expectation. Thank you.

---

### Official Review · Reviewer_Hh1F · 2021-11-06

**Correctness:** 4
**Technical Novelty And Significance:** 3
**Empirical Novelty And Significance:** 3
**Recommendation:** 6
**Confidence:** 3

**Main Review:**

The modeling of this differentiable physics paper is very fine-grained compare to prior papers. Their contact dynamics can even model the Strobeck effect and also simulate self-excited vibrations. I would very much like to see those detailed dynamics in their video. (By the way, the authors mentioned the supplementary video in the paper several times but I could not find the supplementary materials. Did the authors upload it? Or is there something wrong with my reviewer console?)
The authors also conduct extensive experiments to support their claims.

I also have some concerns after reading the paper,
1. The fine-grained model is accurate. I wonder what its time performance is. Could the author report the wall clock time? I also notice the time step is relatively small (0.001s). What is the biggest time step the simulation can deal with?
2. How this method can scale to some more complex topology? All the experiments are on one square sheet. Can your method simulate something like a T-shirt or dress, which is probably more relevant to design and fabrication?
3. Could the author release the code and video as supplementary materials?


**Summary Of The Paper:**

This paper proposed differentiable yarn-level dynamics for fabrics. Compared to the previous differentiable cloth model, this paper is more specific and fine-grained. They carefully design the shear force and Coulomb friction force so as to keep the physics accurate as well as differentiable. The experiments show that their method can learn the control policy and material parameters successfully.

**Summary Of The Review:**

This paper presents a more fine-grained model compared to other differentiable physics papers. Claims are well supported by the experiments. It would be better if the author can better demonstrate their application to more complex scenarios (like a T-shirt or dress instead of a single square sheet).

---

> ### Author Response · Authors · 2021-11-18
> **Time performance, complex topology, and sharing code.**
>
> Thanks for your comments.
>
> # Wall Clock Time and Maximum Time Step Size
> The simulation wall clock time of cloths in different sizes is given below:
>
> | $5 \times 5$ |  $10 \times 10$ |  $17 \times 17$ |  $25 \times 25$ |
> | --------------  |  --------------  |  --------------  | --------------  |
> | 95 ms/step | 465ms/step | 1302 ms/step | 3400 ms/step |
>
> Note that our implementation is currently a simple CPU-based one with OpenMP. This is mainly to prove the feasibility of the fine-grained differentiable fabric model. A GPU-based implementation will be much faster. We will add it into the future work.
>
> The step size is mainly to do with forward simulation stability and does not affect learning the parameters or the control. The specific step size 0.001s is what we used for generating the ground-truth data via the indifferentiable yarn-level cloth simulator (Cirio et al., 2014). To ensure stability, we used a small step size. After learning the parameters, the step size can be increased as we employ an implicit stepping scheme (Baraff & Witkin, 1998). However, we did not push the system to see the largest possible step size, which is related to many things, such as the stiffness of the material, the external forces, etc. We will add discussions to explain this.
>
> # Complex Topology
>
> The experiment design is following existing protocols in textile, not arbitrarily decided by us. Our experiment setting is inspired by KES-FS which measures cloth physical parameters on small cloth samples instead of an entire garment (Kawabata, 1980).  A similar setting is also employed in (Liang et al, 2019). In addition, our main objective is proposing a fully differentiable cloth simulator with differentiable force models. In theory, the parameters learned on small cloth can be used to simulate cloth in any size or topology. The parameters can be even fed into other yarn-level simulators for simulation.  We will add simulation with more complex topologies into the future work.
>
> # Videos and Code
> URL links of videos were given in the Appendix. We also give them here:
>
> Evaluation experiments: [https://youtu.be/HfflAbSaVaw](https://youtu.be/HfflAbSaVaw)
> Comparison experiments: [https://youtu.be/Tq5H8qb2-90](https://youtu.be/Tq5H8qb2-90)
> Simulation with collisions and larger cloths: [https://youtu.be/n5mfHe0Lmo4](https://youtu.be/n5mfHe0Lmo4)
> Error accumulation tests: [https://youtu.be/r7zQVcYGr2E](https://youtu.be/r7zQVcYGr2E)
>
> We will share our code upon acceptance.
>
> ## Additional References
> Kawabatra, S., 1980. The Standardization and Analysis of Hand Evaluation. *The Textile Machinery Society Japan*.

---

### Decision · Program_Chairs · 2022-01-20

**Decision:**

Accept (Poster)

**Comment:**

This paper introduces a differentiable yarn-level model of fabrics. The model is more detailed and physically realistic than proposed in earlier work, which may allow for applications to manufacturing guidance and textile design.
The paper is generally well-written and contains detailed problem formulation and derivations.
Experiments show it is possible to successfully learn a control policy and material parameters using the differentiable model.